# IMU Airtime Detection in Snowboard Halfpipe: U-Net Deep Learning Approach Outperforms Traditional Threshold Algorithms

**DOI:** 10.3390/s24216773

**Published:** 2024-10-22

**Authors:** Tom Gorges, Padraig Davidson, Myriam Boeschen, Andreas Hotho, Christian Merz

**Affiliations:** 1Research Group Snowboard, Department Strength, Power and Technical Sports, Institute for Applied Training Science, 04109 Leipzig, Germany; myriam.boeschen@medizin.uni-leipzig.de (M.B.); merz@iat.uni-leipzig.de (C.M.); 2Chair for Data Science, Center for Artificial Intelligence and Data Science, University of Würzburg, 97074 Wuerzburg, Germany; davidson@informatik.uni-wuerzburg.de (P.D.); hotho@informatik.uni-wuerzburg.de (A.H.); 3Institute of Biomechanics and Orthopaedics, German Sport University, 50933 Cologne, Germany

**Keywords:** event detection, freestyle sports, binary segmentation, airtime, convolutional neural network, elite athletes

## Abstract

Airtime is crucial for high-rotation tricks in snowboard halfpipe performance, significantly impacting trick difficulty, the primary judging criterion. This study aims to enhance the detection of take-off and landing events using inertial measurement unit (IMU) data in conjunction with machine learning algorithms since manual video-based methods are too time-consuming. Eight elite German National Team snowboarders performed 626 halfpipe tricks, recorded by two IMUs at the lateral lower legs and a video camera. The IMU data, synchronized with video, were labeled manually and segmented for analysis. Utilizing a 1D U-Net convolutional neural network (CNN), we achieved superior performance in all of our experiments, establishing new benchmarks for this binary segmentation task. In our extensive experiments, we achieved an 80.34% lower mean Hausdorff distance for unseen runs compared with the threshold approach when placed solely on the left lower leg. Using both left and right IMUs further improved performance (83.37% lower mean Hausdorff). For data from an algorithm-unknown athlete (Zero-Shot segmentation), the U-Net outperformed the threshold algorithm by 67.58%, and fine-tuning on athlete-specific (Few-Shot segmentation) runs improved the lower mean Hausdorff to 78.68%. The fine-tuned model detected takeoffs with median deviations of 0.008 s (IQR 0.030 s), landing deviations of 0.005 s (IQR 0.020 s), and airtime deviations of 0.000 s (IQR 0.027 s). These advancements facilitate real-time feedback and detailed biomechanical analysis, enhancing performance and trick execution, particularly during critical events, such as take-off and landing, where precise time-domain localization is crucial for providing accurate feedback to coaches and athletes.

## 1. Introduction

In the Olympic discipline of snowboard halfpipe, the athletes perform tricks by pushing through the transition, taking off at the coping, using the airtime for tricks, and landing at the identical wall. This performance is scored subjectively by judges whose scores are based on criteria established by the International Ski and Snowboard Federation (FIS), including trick difficulty, amplitude, variety, and execution [1]. It can be observed that the trick difficulty, airtime, and variety have steadily increased over the past years in elite-level competitions [2]. For the judging criteria amplitude, the traveled horizontal distance should be in proportion to the vertical amplitude. It is self-evident that a large amplitude results in a long airtime. The difference between amplitude and airtime is that a deep landing (large distance between the landing position and the coping in the vertical direction) extends the airtime, but has no effect on the amplitude [1]. Therefore, the time over coping is more relevant for the judging criteria ’amplitude’ as the airtime, which describes exactly how long the rider has time in the air to perform a trick. Nevertheless, airtime is an important performance parameter [3], a prerequisite for tricks with a high amount of rotations, and provides a certain amount of information about the amplitude. The number of rotations in turn has a major influence on the difficulty of the tricks, which is the most important judging criteria [4]. Besides airtime, information on movements at the airtime-related events ’take-off’ and ’landing’ can give riders feedback and enable in-depth biomechanical analyses, as shown by Thelen et al. [5], offering valuable insights to enhance athletes’ performances. If the events take-off and landing are known, data recorded in between can be assigned to the airtime or accordingly the riding phase. Consequently, the data can be used further to describe and objectify the movement during take-off, airtime, landing, and riding. To offer this analysis in everyday training, for example, as a real-time feedback system, as shown by Thelen et al. [5], the event detection needs to be automatically due to time constrains. Beyond that, it is also necessary to detect the take-offs and landings to develop and apply trick classification algorithms in the future. However, the current methods of determining airtime and the corresponding events of take-offs and landings primarily involve manual video file assessments [2,6]. This process is not only time-consuming but also contingent on the quality of the available videos, especially the recording frame rates [7], making it potentially incomplete or inaccurate.

A promising approach to improve airtime detection is the use of inertial measurement units (IMUs), which have already been widely used for human activity recognition (HAR) tasks [8,9,10]. Airtime detection algorithms based on IMU data in particular have already been introduced by several authors [11,12]. Such algorithms are also previously applied in commercial products (e.g., Garmin MTB metrics (Garmin Ltd., Schaffhausen, Switzerland) or Swiss Timing data for live data in television broadcasts (Swiss Timing Ltd., Corgémont, Switzerland)), but their accuracy, reliability, and exact implementation are not known yet.

Snowboard freestyle-specific methods found in the literature rely on a threshold-based algorithm [13] and a probabilistic approach using multiple attribute decision-making to decide on probabilities of detected acceleration peaks based on extracted features like amplitude or proximity to other peaks [3]. Machine learning (ML) algorithms, with their capability to process and analyze complex data sets, offer promising avenues to address the limitations of current threshold-based methods and have already found wide application in HAR and event detection [8,9,10]. Therefore, this paper aims to explore the potential of ML in enhancing the accuracy and reliability of detecting and analyzing airtime, take-off, and landing events. Additionally, it investigates the significance of the volume of sensor data for this task.

Therefore, we hypothesize that machine learning models, trained on robust datasets and fine-tuned to the specific nuances of IMU data in snowboard freestyle, can significantly enhance the precision of data capture and reduce the need for manual data processing. In this context, the present work compares the traditional threshold-based algorithm with a fully supervised convolutional neural network (CNN) approach, specifically a U-Net architecture adapted from Zhang et al. [14], focusing on event detection with IMU data from snowboard halfpipe runs in an elite training setting.

## 2. Related Work

### 2.1. Scientific Approaches to Quantify Airtime in Snowboarding

There are several algorithms to identify jumps in snow sports without detecting airtime. Kranzinger et al. [15] pursued a threshold-based approach using ski boot-mounted IMUs, Roberts-Thomson et al. [16] utilized a fuzzy logic approach with smartphone IMU data, and Sadi and Klukas [17] employed a cross-correlation approach with head-mounted IMUs. However, apart from the detection of big air jumps by Kranzinger et al. [15] (100% accuracy), these approaches showed errors of 6–8% and even 56% for jumps lasting less than 500 ms with Kranzinger et al. [15]. In Snowboard Freestyle contests, not only the occurrence of a jump itself but also the related airtime is a key judge criterion [1] and an important performance variable [3]. There are different approaches to detect airtime-relevant events. For kicker jumps and grinds, Groh et al. [18] used the accelerometer data of an inertial-magnetic measurement unit (IMMU) fixed on the snowboard combined with a threshold approach. Sadi et al. [3] and Lee et al. [19] used head-mounted micro-electro-mechanical systems IMU combined with a probabilistic approach using multiple attribute decision-making to determine the airtime of snowboard jumps. Friedl et al. [20] developed a way to detect single grab events based on peak detection of jerk data calculated by the acceleration data using an IMU mounted on the board. For this application, it is essential to distinguish between the IMU data recorded during airtime and the data collected while riding in contact with the snow. Snowboard airtime detection for halfpipe runs using a basic threshold algorithm on accelerometer raw data of an IMU fixed on the lower back has already been proposed by Harding et al. [21] and will be used as a baseline in the following.

### 2.2. Deep Learning in Sports Sciences

Deep learning has gained attention in recent years as sensors are easily usable during sports exercises. Furthermore, the fast and objective evaluation of senor data is highly beneficial for direct feedback and improving training outcomes. The application of deep learning models in sports is extremely diverse and wide-ranging from, e.g., health tracking, like estimating runner fatigue using neural networks [22,23] to tactical analysis in football using AI assistants [24]. Also, its application in snow sport airtime data has already resulted in a promising approach predicting ski jump length [25].

## 3. Materials and Methods

In this section, the subjects, measurement system, and used procedures are presented in chronological order.

### 3.1. Subjects

Eight elite snowboard freestyle riders (2 ♀, 6 ♂; age: 18.4±3.3 years; mass including sports equipment: 73.9±6.8 kg; height: 172.6±9.3 cm; 4 regular, 4 goofy stance) of the German National Team performed 626 snowboard halfpipe tricks (example see Figure 1) in a competition-ready superpipe (Kitzsteinhorn, Austria: $S_{1}$ and $S_{3}$ and Laax, Switzerland: $S_{2}$–$S_{8}$). Riders performed 1–10 hits per run characterized by 1.24±0.19 s (max: 1.91 s; min: 0.4 s) airtime and 0°–1080° rotations according to their trick name (see Table 1). Only runs with clear video evidence of a successful landing were included in the analysis. The study was conducted in accordance with the Declaration of Helsinki and procedures were approved by the Regional Ethics Committee (number of approval: 214/2022). Informed consent was obtained from all riders prior to their participation in the study.

### 3.2. Measurement System

Movements were recorded by two IMUs (Shimmer3 IMU Unit, Shimmer Wearable Sensor Technology, Dublin, Irland) at 201.03 Hz and filmed by a video camera (Panasonic HC-X1500E, Panasonic, Kadoma, Japan) at 100 Hz. Due to the different sampling frequencies, the accuracy of timing data depends on the video recording frame rates. The IMU devices were strapped to the lateral side of both boots above the ankle strap. Data were stored on an internal storage.

### 3.3. Procedures

The IMUs were synchronized manually to the video by filming multiple taps on the resting sensor before riders dropped into the halfpipe. This leads to clearly distinguishable events in both the video and the IMU data, which are used to align both data sources in the time domain [26,27,28]. The event take-off (Figure 1A—last contact of the snowboard with the halfpipe before the trick [6]) and landing (Figure 1C—first contact of the snowboard with the halfpipe after the trick [6]) were detected manually from videos and upscaled to 201.03 Hz. The IMU data were labeled (see ground truth Figure 1) with the help of manual event detection leading to each frame being binary coded: 1 (in the air) and 0 (contact to snow).

### 3.4. Methodology

Detecting transitions between take-off and landing in the context of a snowboard halfpipe jump can be seen as a change point detection (CPD) or a binary segmentation task. However, the change points from the binary segmentation mask can be obtained by identifying the transitions from 0 (on the ground) to 1 (in the air).

More formally, we want to annotate the data obtained from the IMU $x\in R^{d\times T}$ into binary labels: $F\left(x\right)=R^{d\times T}\mapsto B^{T}$, where *d* is the number of features used from the IMU, and *T* is the length of a single run or window. A neural network realizes this mapping function F(x).

### 3.5. Neural Network

In our studies, we utilized a 1D CNN U-Net architecture. The U-Net architecture has already been used successfully for segmentation tasks on images [14,29,30], while the 1D CNN achieves state-of-the-art results in analyzing temporal data for classification tasks [22,31]. The U-Net structure is characterized by its encoder-decoder pathway, featuring convolutional blocks for deep feature extraction and transposed convolutions for precise localization with residual connections on the same depth. The architecture comprises an encoder for downsampling, a bottleneck, and a decoder for upsampling. Each section uses convolutional operations to manipulate the input data, refine features, and reconstruct the output with detailed segmentation. Compare Figure 1 in Zhang et al. [14] for a visualization of the architecture.

Encoder: The encoder path consists of a series of downsampling blocks. Each block applies two 1D convolutional layers followed by a ReLU activation:
(1)$$x_{\mathrm{conv}}=\mathrm{ReLU}\left(\mathrm{Conv}1D\left(x\right)\right),$$
followed by max pooling and dropout for downsampling and regularization:
(2)$$x_{\mathrm{down}}=\mathrm{Dropout}\left(\mathrm{MaxPool}1D\left(x_{\mathrm{conv}}\right)\right).$$Bottleneck: The bottleneck, at the deepest level, applies a double convolution block without downsampling, processing the most compressed feature representations:
(3)$$x_{\mathrm{bottleneck}}=\mathrm{DoubleConvBlock}\left(x_{\mathrm{last}\_\mathrm{downsampled}}\right).$$Decoder: The decoder path employs transpose convolutions for upsampling, followed by concatenation with the corresponding encoder feature map and a double convolution block:
(4)$$\begin{array}{cc} x_{\mathrm{up}} & =\mathrm{Conv}1\mathrm{DTranspose}(x_{\mathrm{prev}}), \end{array}$$
(5)$$\begin{array}{cc} x_{\mathrm{concat}} & =\mathrm{concatenate}(\left[x_{\mathrm{up}},x_{\mathrm{last}\_\mathrm{downsampled}}\right]), \end{array}$$
(6)$$\begin{array}{cc} x_{\mathrm{final}} & =\mathrm{DoubleConvBlock}(x_{\mathrm{concat}}). \end{array}$$

The output of the final upsampling stage is passed through a Conv1D layer without activation and a kernel size of one to produce the logits of the segmentation mask:(7)$$\hat{y}=\mathrm{Conv}1D\left(x_{\mathrm{final}}\right).$$

#### Implementation Details

The model is realized using the TensorFlow Keras API [32], optimized with Adam [33], and trained using binary cross-entropy to suit binary segmentation tasks. We used a fixed batch size of 256 to fully utilize the RTX 2080 GPU (NVIDIA Corporation, Santa Clara, CA, USA). To prevent overfitting, an early stopping algorithm (patience 5 epochs) was employed, incorporating binary intersection over union (IoU) as a secondary metric. This metric is often used to measure success in segmentation tasks, and is particularly valuable because it focuses on the overlap between predicted and true segments, ignoring true negative predictions [34]. The performance metric (see Section 3.8), was not utilized as a stopping criterion, as it can not be efficiently calculated on the GPU, lowering throughput in the training process.

### 3.6. Baseline

The stated classical threshold approach [21] will be used in this paper as a reference. The methodology devised by Harding et al. [21] is centered around a two-pass process tailored for analyzing half-pipe snowboarding activities. The first stage involves the identification of snowboard run locations, achieved by assessing the power density within the frequency domain of the raw IMU signals. The second stage focuses on calculating the duration of air-times for various aerial acrobatic maneuvers. This is done through a time-domain search algorithm that operates based on predefined thresholds and is followed by an exclusion procedure for unrealistic air-times, which, according to Harding et al. [21], would be outside 0.8 s and 2.2 s. The threshold divides the raw data into high (1) and low (0) states and is defined by:(8)$$\mathrm{High}=\mathrm{Data}\geq \left(\max\left(\mathrm{Data}\right)+\mathrm{mean}\left(\mathrm{Data}\right)\right)\times T_{h}$$
where $T_{h}$ is the high state threshold, which was determined experimentally with a value of 0.25 based on the authors’ data. A mentioned transition state was not used by the authors for further calculations, and therefore, the airtime start was defined as the transition from state low to state high and landing was defined as the transition from state high to state low. Low is, in this scenario, interpreted as:(9)Low=Data≤High

This dual-stage approach proved exceptionally efficient, managing to accurately determine air-times in all instances, which included a total of 92 maneuvers performed by four different athletes. Furthermore, the method’s reliability is highlighted by its strong correlation (r=0.78±0.08) with air-time measurements derived from video-based reference methods. The statistical significance of these results is reinforced by a *p*-value below 0.0001. While the method displayed a minor mean bias of −0.03±0.02 s, improvements in accuracy are anticipated with the adoption of the machine learning approach presented in this work.

### 3.7. Dataset Splits

Due to the different fixation points of the IMUs, we outline different experimental settings to demonstrate the efficacy of our approach. The source code for all our experiments will be available upon acceptance.

#### 3.7.1. Splitting by Run

For this setup, we used data obtained from subjects S_1_ to S_7_, using only the left IMU. Left, in this case, refers to the sensor being positioned on the left side of the sagittal plane from the subject’s perspective. For goofy riders, the sensor is at the back in their normal stance, while for regular riders, it is at the front. The relationship between “left” and “right” can change relative to “front” and “rear”, as athletes are able to switch riding directions during a run. We created a train/validate/test split by using all but two full runs of each athlete in the training, one of each in validation, and one of each in the test set (leave-one-run-out). A split by the athlete to implement a leave-one-subject-out cross-validation was not possible due to the small sample sizes from the two measurement locations. However, in the chosen setup, we can not only present the efficacy of our approach but also test the amount of labels that are necessary for a high-quality segmentation. For this, we ran separate hyper-parameter studies with weights and biases [35] by using 20%, 50% and 100% of available windows with 100 sweeps in each setup. Windows were created by a sliding window approach with a stride of 5 frames and a window size of 400 frames. They were further normalized to units of g by dividing the raw acceleration by 9.81 m/s^2^. The window size was chosen as 400 frames for the following reason: considering the capture frequency of 201.03 Hz, a window size of 400 frames will create data windows, including the complete jump featuring take-off and landing, even for the maximum airtime of 1.91 s. The range and distribution of the hyper-parameters are available in Table 2.

#### 3.7.2. Predictions on an Unseen Athlete

For this setup, we use data from athlete S_8_ as out-of-distribution (leave-one-runner-out) data. After training the optimal model in Section 3.7.1, we use it to create predictions for all but one run of S_8_. We discuss each of these runs of the athletes independently for their respective metrics (see Section 3.8). This setup demonstrates the Zero-Shot (ZSL) capabilities of our approach.

#### 3.7.3. Finetuning on New Athlete

Similarly as above, we tested one more setup by using the left out run of $S_{8}$ from Section 3.7.2 to fine-tune (5 additional epochs of training) the best model on the specifics of the athlete. These results were compared with the above to outline if a finetuning step was beneficial to capture these characteristics, as they might be obtained on a warm-up run in each competition. This setup demonstrates the Few-Shot (FSL) capabilities of our approach.

#### 3.7.4. IMUs at the Front and Rear Foot

Subjects S_1_ to S_7_ had IMUs located at both the front and the rear foot. To check whether the use of both data sources offers added value for the algorithm, which can be assumed based on Gorges et al. [36], a hyper-parameter tuning was also carried out on the bilateral dataset, and the best model was used to predict the take-off and landing events, as well as the resulting air-times of individual test runs.

### 3.8. Metrics

The main metric used for optimization and assessment of the proposed algorithms was the Hausdorff distance, which measures the extent to which two subsets, A and B, of a metric space differ by determining the maximum distance one must travel from a point in A to the closest point in B [37], leading to a possible range from 0 (perfect) to 400 (window size; worst). This distance is often employed to evaluate the performance of automatic segmentation methods, as it effectively indicates the largest segmentation error [38]. In this study, the average Hausdorff distance per data window was used as a metric. It is important to note that datasets with many windows lacking changepoints tend to achieve lower average Hausdorff values more easily, which is why it is of particular interest to look at the changes within the data of individual athletes.

## 4. Results

In this section, we describe the results of our different dataset splits with their depicted experiments. Exemplary raw predictions for individual runs are presented in Appendix B. The metrics reported are averaged Hausdorff values per window.

The first subsection demonstrates the efficacy of our approach, including an ablation study for the number of required labels and the number of IMU sensors. The succeeding subsection presents the results when using the full halfpipe run as one window contrary to the fixed 400 frame windows. The last subsection showcases the usability of our approach in transfer learning settings.

### 4.1. Splitting by Run

Table 3 displays the results of our experiments when splitting by run. The upper half of the table contains the Hausdorff values when using only the left IMU sensor, while the lower half shows these values when including the front and rear foot in the analysis. Each half was split by three to include the ablation study for the number of labels used for training. Bear in mind that the validation and test sets in each setup were identical for a fair comparison.

#### 4.1.1. Number of IMUs

As shown in Table 3, the use of both IMUs yields to better performances, both in validation and test (6.422 vs. 5.820). Therefore, the right foot seems to contain additional information about ground contact and lift-off, yielding a more precise decision for the start of the jump. However, the difference in validation and test prediction is higher when using both IMUs (4.457/6.422 vs. 3.295/5.820), suggesting stronger overfitting compared to the left IMU-only setting, though detailed analysis on the specific windows with erroneous predictions is required. The optimal hyperparameters for the left IMU were found to be a dropout rate of 0.1177, a learning rate of 0.001146, and 32 units. For the left/right IMU setup, the optimal parameters were a dropout rate of 0.1677, a learning rate of 0.0005822, and 32 units. An overview of the hyperparameter tuning for the different setups is provided in Appendix A.

Especially when comparing with the threshold-based baseline (see Figure 2), our approach performs better by one order of magnitude (27.524 for validation and 34.337 for test). The described threshold algorithm (Section 3.6) was optimized for $T_{h}$ in a range from −1 to 1 in increments of 0.05 on the left IMU x-axis data. The best Hausdorff performance was found for $T_{h}=0.3$, which is almost equal to the $T_{h}$ of 0.25 determined by Harding et al. [21].

#### 4.1.2. Amount of Training Samples

Table 3 also contains rows for various thresholds of available training samples in each setup. Same as in the previous section, we retained the identical validation and test sets at all levels. When using only the left IMU, a quick decay in the validation Hausdorff distance was observed (4.457 vs. 4.986 vs. 5.890), while the difference in test sets remained constant at 0.8 (6.6422 vs. 7.286 vs. 8.087). This suggests overfitting in the training process and indicates that a critical amount of labels (or a variety of characteristics) is necessary.

When using both IMUs, metrics also decrease in setups with fewer training labels (5.820 vs. 7.762 vs. 8.149). In contrast to the left IMU-only setting, there are two key differences: the gap from validation to test grows more quickly with the amount of labels, and it only performs better when using all available labels. The larger input dimensionality (6 vs. 3) seems to require more training samples for discriminating features within the network, explaining the better performance and the stronger tendency of overfitting.

### 4.2. Predictions on Single Runs

In the previous sections, we outlined the results for the windowed input samples, mainly used to augment the number of labeled samples. To validate the efficacy of our approach for the judgment of full trials, we discuss the results in this setup. The binary segmentation mask of the full trial was obtained by averaging duplicated timestamps of overlapping windows yielded by the network. Since the baseline is invariant to the windowing approach, thresholds were directly applied to the full trial in this setting. We used the final model found in the left IMU-only setting with all available labels.

#### 4.2.1. Hausdorff Domain

Table 4 lists the Hausdorff metric on the single test trial of athletes S_1_–S_7_. Similar to the windowed setting, our approach outperforms the baseline by a large margin. The second column in the table shows the Hausdorff metric when using only the left IMU, the next column when using both IMUs and the last column represents the threshold approach on the left IMU. All approaches show the worst metric values with runners S_2_ and S_3_, who exhibited the longest average airtimes, indicating that their jumping characteristics are badly captured via simple thresholding, while our approach shows greater improvements in these settings. Same as in the windowed setting, using both IMUs improves metrics except for S_6_ and S_7_. Here, using a single IMU showed better results.

#### 4.2.2. Time Domain

The Hausdorff metric reports the worst placement of a single changepoint in the number or frames. However, since we are interested in air-time as a unit of time, we additionally analyzed the conversion into the time domain. We visualized these predictions for our approach in Figure 3 for each of the three states (take-off, landing, airtime) independently. Recall that take-off in this plot represents the time difference in a change from 0 to 1, while landing represents the time difference in a change from 1 to 0. Airtime is the state of consecutive ones, incorporating errors predicted in the other two states.

Since the threshold algorithm predicted too many air-times ((204.97±32.54)% of ground truth air-times), time domain differences of predictions are only reported for our approach (100.00% of ground truth air-times).

Median deviations stayed all within ±0.005 s (which is equal to one frame offset at 201.03 Hz) while the interquartile ranges (IQRs) decreased for the take-off differences from 0.025 s (left IMU) to 0.020 s (left + right IMU) for the landing differences from 0.019 s (left IMU) to 0.015 s (left + right IMU) and for the resulting airtime differences from 0.025 s (left IMU) to 0.024 s (left + right IMU).

As illustrated in Figure 3, using both IMUs especially improves predictions for the landing. This decrease is further supported by the reduction in the magnitude and number of outliers.

### 4.3. Transfer Learning

In the previous sections, we employed leave-one-trial-out splits for each runner, which allowed us to demonstrate the efficacy and improvements of our approach in comparison to the threshold-based algorithm. Even though this is useful in many applications, we are also interested in the transfer learning (TFL) capabilities of our approach, i.e., we do not want to show the model the jumping styles for each runner.

In Table 5, the results on different trials from S_8_ are presented, who was not part of any previous split. In the ZSL (zero-shot learning) column, we depict the predictions on these runs (4) with the final model obtained in Section 3.7.1 without any training. In the FSL (few-shot learning) column, predictions are provided after fine-tuning this model with samples from a single run for 5 epochs. To show that the model did not suffer from the catastrophic forgetting phenomenon [39], we included 5-epoch training from scratch. The last column shows the Hausdorff metric for the baseline approach.

We can see that our approach outperforms the threshold algorithm even in the ZSL setup, which did not contain any fine-tuning (32.71 vs. 100.90 on average). Results are further improved by incorporating data from this athlete (FSL) (32.71 vs. 21.51 on average). This setup is useful to have a pre-run available from athletes before the competition and can thus learn characteristics from the jumper and the environment. The time required for fine-tuning with the described setup was 46 s. For comparison, the best models trained for Section 3.7.1 on 100% data took 688 s (left) and 1823 s (left + right) to train. Training a new model from scratch (with the optimal parameters) yields worse performance than the baseline for this athlete (304.10 vs. 100.90 on average).

The increase in accuracy due to the fine-tuning is also evident when looking at the deviations of predictions from the ground truth in the time domain (Figure 4). Take-off deviations decreased from a median of 0.012 s (IQR 0.037 s) to 0.008 s (IQR 0.030 s), landing deviations from −0.015 s (IQR 0.052 s) to 0.005 s (IQR 0.020 s), and the resulting airtime deviations from a median of −0.055 s (IQR 0.128 s) to 0.000 s (IQR 0.027 s). This decrease is further supported by the reduction in the magnitude and number of outliers.

As we can see from run 17 (Figure A8), our approach shows greater Hausdorff values in specific scenarios with outliers in the raw sensor data due to external impacts (51.64 vs. 32.71 on average), while the baseline performs equally in comparison to different runs (104.06 vs. 100.90 on average).

## 5. Discussion

This study aimed to enhance the detection of take-off and landing events in snowboard freestyle using inertial measurement unit data in conjunction with machine learning algorithms. Overall, the results demonstrate that the transition from traditional threshold-based algorithms to machine learning approaches and sensor fusion is very promising. Concerning a simple binary jump/no jump detection, with an accuracy of 100%, our approach was already better than previous traditional approaches using cross-correlation (error: 8%) [17], fuzzy logic (error: 8%) [16], or threshold (error: 0% big air, 6% medium jumps, 56% small jumps) [15], which only aimed at the pure recognition of jumps in snow sports and not their precise duration.

Impact of Dual-Sensor Setup and Expanded Datasets on Temporal Event Detection

In the present study, with a focus on precise temporal event detection, the conducted hyperparameter tuning resulted in a 9.37% lower mean test Hausdorff distance for the configuration that incorporated data from two sensor positions at both feet compared to a single sensor setup. This supports the assumption of Gorges et al. [36] that the acceleration characteristics of the left and right foot differ in a manner that provides additional valuable information when both sources are considered. These differences are related to the varying riding directions (normal/switch), stance (regular/goofy), and the non-rigid characteristics of the board. The mean improvements by the dual sensor setup were not found for S_6_ and S_7_, which are the least experienced athletes who consequently showed the shortest airtimes. In future applications, it is anticipated that algorithms might perform even better if the riding direction is included as a feature at each time point since a clear allocation of the front and rear foot is possible. This could also benefit the single-sensor setup, as the left sensor may alternate between the front and back depending on the athlete’s stance and riding direction. The fact that the determined optimal hyperparameters do not represent extreme values within the specified parameter range confirms the appropriateness of the chosen ranges for tuning and suggests that the models are well-optimized based on the given data. The data reduction analysis demonstrated that expanding the dataset for training U-net algorithms enhances accuracy, evidenced by lower Hausdorff values for the test data in event detection. This improvement is observed both when considering data from only the left-sided IMU and when utilizing data from both sides. The enhanced performance can be attributed to the availability of a broader spectrum of sensor data curves during the learning process, indicating a high diversity of characteristic landing and take-off acceleration curves. Consequently, for future applications, an even larger dataset would be desirable. Additionally, incorporating sensor inputs like gyroscope data from the IMUs should be explored for potential performance improvements. Such an expanded dataset also has the potential to capture a wider array of variations specific to different subjects, tricks, conditions, or locations, thereby improving the segmentation of acceleration curves.

Threshold Algorithm Evaluation

The threshold algorithm, which serves as a baseline in this study, showed a $T_{h}$ multiplier when optimized on the present dataset, which differed from Harding’s optimized value by only 0.05. This suggests that the data foundation, despite different boundary conditions (rider, location, tricks, sensor position), contains similar characteristics and that Harding’s approach is transferable to external data. However, the magnitude of the Hausdorff values for the threshold algorithm compared to the U-net approaches (left/right) differ by 82.61%, which indicates potential for improvement in segmentation through machine learning. Additionally, since the threshold algorithm in our case, unlike Harding’s, was not even able to consistently identify the correct number of jumps (usually detecting too many), its practical applicability must be questioned.

U-Net Superiority and the Option for Balanced Fine-Tuning

In contrast to the threshold algorithm, the proposed U-Net approach demonstrated significantly better performance and was able to accurately identify every jump for known athletes. For the unknown athlete (S_8_), one false positives (Run 17; Table 5; Figure A8) was detected. However, this can be corrected in the future with simple post-processing algorithms based on known parameters such as realistic jump durations, periodicity of jumps, or minimum riding times between individual jumps. Since the U-Net prediction for the runs of the unknown athlete also significantly outperformed the threshold algorithm, it demonstrates the transferability of a trained model with the chosen approach. The further improvements achieved through fine-tuning in all tested instances indicate the model to lack sufficient comprehensive data to capture all new characteristics. This suggests that a larger model has the potential to enhance robust airtime detection on new datasets. However, excessive robustness might compromise precision. Therefore, a balanced compromise between data volume and specific fine-tuning, such as athlete- or location-specific adjustments, should be pursued in the future. Since there are often training runs and qualification runs, and considering the relatively short amount of time required for a fine-tuning (< 1 min), it would be feasible to input corresponding datasets before a competition. It was also ruled out that simply training from scratch according to the finetuning setting (only 5 epochs) is sufficient (see Table 5).

The IQR values and the distribution of the outliers (Figure 3 and Figure 4) showed that landings can be detected more accurately than take-offs. This can be explained by clearer impacts during the landing compared to the smoother take-off characteristics.

Limitations, Applicability, and Outlook

One limitation of this study is the manual sync of sensors and video images as well as manual labeling, which might cause inaccuracies. Concerning the chosen window size of 400 frames at 201.03 Hz, it can be excluded that only airtime frames are present within a window, even for the longest airtimes in the dataset. However, the limited number of frames with ground contact appears to make it more challenging for the algorithm, as indicated by the higher Hausdorff values for subjects with the longest airtimes (Table 1 and Table 4). This issue might have an even stronger impact when analyzing world-class airtimes exceeding 2 s [2]. Therefore, optimizing the window size for practical use cases, especially for world-class athletes with extended airtimes, is crucial for enhancing the algorithm’s accuracy and reliability. The possibility of transferring the results to other freestyle disciplines and appropriate fine-tuning must be examined in follow-up studies. For these follow-up studies, we also advise striving for larger datasets to enable leave-one-subject-out cross-validation approaches, which can help mitigate possible issues arising from athlete heterogeneity. Overall, the approach of determining airtimes using machine learning algorithms on IMU data can be described as an improved method compared to traditional algorithms and should be incorporated into practical applications in the future. The identified median prediction difference to the ground truth of 0.005 s (IQR = 0.024 s) for airtimes of seen athletes would translate to an error of only 0.24% when applied to world-class performances with mean airtimes of 2.1 s [2]. For the model with fine-tuning, the median difference from the prediction to the ground truth showed no deviation at all (difference = 0.00 s; IQR = 0.027 s). Therefore, the magnitude of these errors falls within a range that allows for further analysis with satisfactory accuracy.

## 6. Conclusions

This study successfully enhanced the detection of take-off and landing events in snowboard freestyle using IMU data and machine learning compared to a traditional threshold approach. The U-Net convolutional neural network (CNN) significantly outperformed traditional threshold-based methods, achieving up to 83.37% lower mean Hausdorff distances and demonstrating high precision in predicting take-off, landing, and resulting airtime events. Utilizing both left and right IMUs improved accuracy, highlighting the value of sensor fusion. Athlete-specific fine-tuning further enhanced model performance, indicating the potential for even greater improvements with larger datasets and additional features such as riding direction. The findings suggest practical applications for real-time feedback and biomechanical analysis in snowboarding and other freestyle sports. Future research should focus on optimizing window sizes for longer airtimes and automating the synchronization and labeling process to further refine this methodology. This machine learning-based approach represents a substantial improvement over traditional methods and should be considered for integration into competitive snowboarding.

## Figures and Tables

**Figure 1 sensors-24-06773-f001:**
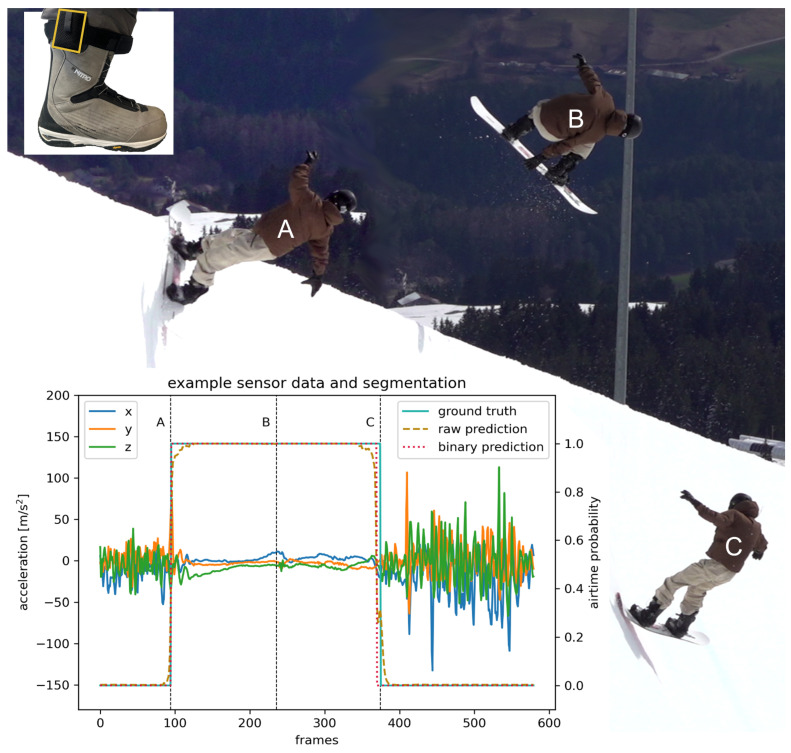
Example jump and detail of IMU (yellow) attachment with associated sensor data and U-Net airtime prediction compared to ground truth, with a particular focus on the events: take-off (A), mid-air (B), and landing (C).

**Figure 2 sensors-24-06773-f002:**
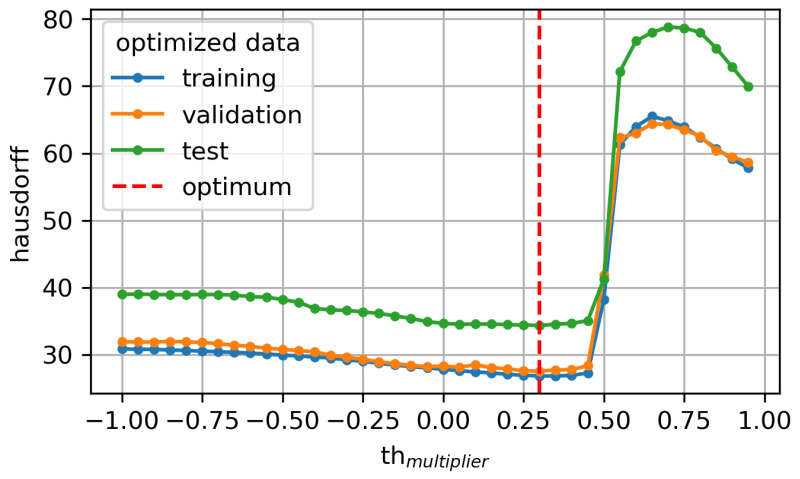
Hausdorff development over a range of multipliers for threshold algorithm optimization with indication of minimal Hausdorff at multiplier = 0.3.

**Figure 3 sensors-24-06773-f003:**
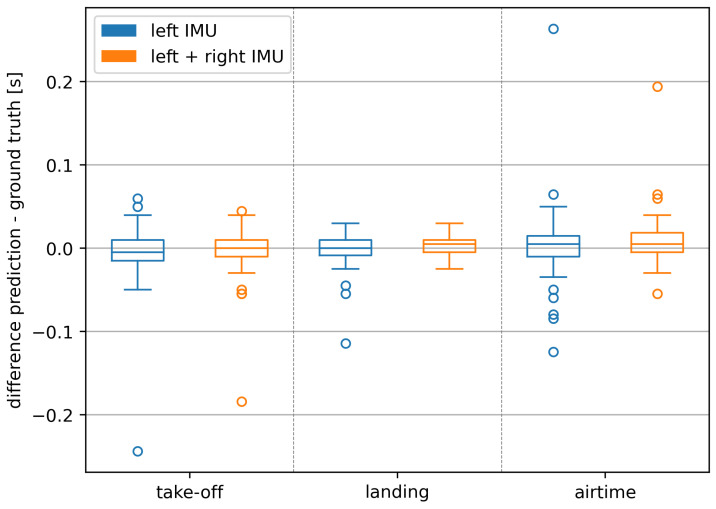
Deviations of predictions for landings, take-offs, and airtimes on runs of seen athletes in seconds.

**Figure 4 sensors-24-06773-f004:**
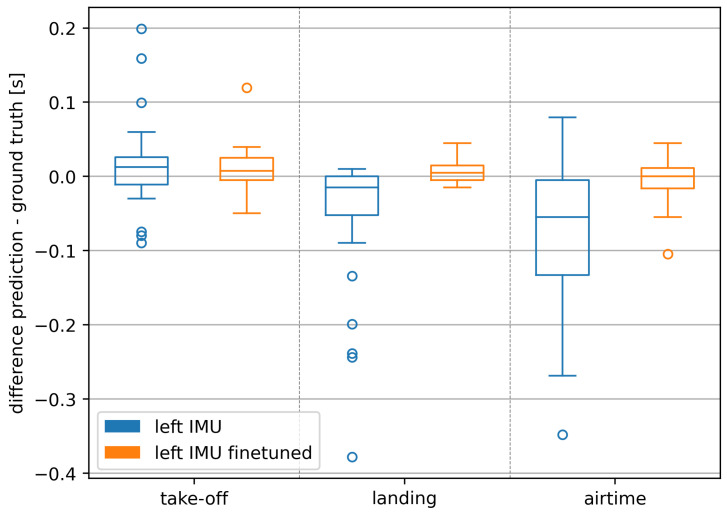
Deviations of predictions for landings, take-offs, and airtimes on runs of a new athlete in seconds.

**Table 1 sensors-24-06773-t001:** Overview of subjects and airtime characteristics.

Subject	Runs	Hits	Airtime [s]
S_1_	9	39	1.11 (14)
S_2_	14	81	1.68 (13)
S_3_	20	82	1.49 (17)
S_4_	25	164	1.10 (27)
S_5_	20	104	1.34 (20)
S_6_	6	35	1.09 (13)
S_7_	13	84	0.85 (22)
S_1–7_	107	589	1.25 (33)
S_8_	6	36	1.26 (12)

**Table 2 sensors-24-06773-t002:** Experiment parameters.

Parameter	Range
lr	$\mathrm{loguniform}(10^{-6},10^{-2})$
dropout	uniform(0.0,0.5)
units	$2^{\mathrm{randint}(2,8)}$
window size	400
window steps	5
data reduction	[1.0,0.5,0.2]

**Table 3 sensors-24-06773-t003:** Results of data reduction.

Setting	Train Samples [%]	Hausdorff_val_	Hausdorff_test_
Left	100	4.457	6.422
	50	4.986	7.286
	20	5.890	8.087
Left/Right	100	3.295	5.820
	50	3.134	7.762
	20	4.054	8.149

**Table 4 sensors-24-06773-t004:** Hausdorff predictions on single test runs of seen athletes.

	Setup		
Test Subject	Left	Left/Right	Baseline
S_1_	7.41	3.89	34.06
S_2_	11.23	9.76	71.84
S_3_	21.57	17.97	53.22
S_4_	2.53	2.14	36.70
S_5_	6.52	3.40	34.40
S_6_	2.93	5.48	25.01
S_7_	2.66	3.74	23.83
Mean	7.84	6.63	39.87
SD	6.34	5.16	15.82

**Table 5 sensors-24-06773-t005:** Hausdorff of predictions on single test runs of an unseen athlete using the left IMU. ZSL refers to zero-shot learning, and FSL refers to few-shot learning.

		Setup		
Run from S_8_	ZSL	FSL	From Scratch	Baseline
12	23.93	15.48	211.59	105.08
13	29.79	16.10	304.26	87.23
14	25.48	16.02	343.77	107.24
17	51.64	38.43	356.79	104.06
Mean	32.71	21.51	304.10	100.90
SD	11.14	9.88	56.81	7.98

## Data Availability

All code and data used in this study is available under https://github.com/LSX-UniWue/AirtimeDetection, accessed on 19 August 2024.

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
