# Peer review of "IMU Airtime Detection in Snowboard Halfpipe: U-Net Deep Learning Approach Outperforms Traditional Threshold Algorithms"

_sensors, 2024, doi:10.3390/s24216773_

Round 1

Reviewer 1 Report

Comments and Suggestions for Authors

This study presents a 1D CNN U-net that detects take-off, landing, and airtime of a snowboard freestyle based on an IMU attached to the foot, and verifies the performance of the model using data collected from eight elite snowboard riders. The data collection/processing and model training/validation methods were clearly explained. In particular, it is considered meaningful that the performance of the model was verified in various aspects including comparison by amount of training data, comparison by number of IMUs (left only vs. left/right), comparison with the baseline, and comparison with the test dataset from seen/unseen athletes.

However, it seems that several aspects below need to be improved.

- If possible, it would be good to show the details of the IMU attachment in Figure 1.

- In Section 3.7.1, it was explained that only the data of the left IMU was used as model input. However, in order to help readers understand, it seems better to clearly explain whether the left foot is located at the front or rear of the snowboard. In addition, if possible, it would be good to add performance comparison results between using IMU data from the front and rear feet (or right and left feet).

- Although the airtime detection performance was evaluated by the averaged error of Hausdorff domain and time domain, it seems that it would be good to add prediction result graphs such as Figure 1 as samples.

- I would like to suggest including the gyroscope data into the model input and comparing the performance by input data in the future study.

- “3.2. Measurement system”  “3.2. Measurement System”

Author Response

Dear reviewer,

We would like to take this opportunity to thank all reviewers for their comprehensive and constructive reviews. We are very pleased that the reviewer explicitly supports our aim to investigate airtime detection via machine learning methods in more detail. The comments to the reviewers' are given below and the changes to the manuscript are highlighted in yellow.

Comments1:  If possible, it would be good to show the details of the IMU attachment in Figure 1.

Response1: We added an image in the upper left corner of Figure 1, showing the sensor position on the foot.

Comment2: In Section 3.7.1, it was explained that only the data of the left IMU was used as model input. However, in order to help readers understand, it seems better to clearly explain whether the left foot is located at the front or rear of the snowboard.

Response2: In section 3.7.1 it was added: “Left in this case refers to the sensor being positioned on the left side of the sagittal plane from the subject’s perspective. For goofy riders, the sensor is at the back in their normal stance, while for regular riders, it is at the front. The relationship between "left" and "right" can change relative to "front" and "rear" as athletes are able to switch riding directions during a run.” (line 208-212)

Comment3: In addition, if possible, it would be good to add performance comparison results between using IMU data from the front and rear feet (or right and left feet).

Response3: We did not split by specific runs, such that the sensor is only at the front or only at the back. We added a notion on this in the discussion and stated that two sensors perform better anyhow and are preferable. (line 391-393)

Comment4: Although the airtime detection performance was evaluated by the averaged error of Hausdorff domain and time domain, it seems that it would be good to add prediction result graphs such as Figure 1 as samples.

Response4: We added some figures in the appendix B highlighting an example for exemplary predictions of airtime probabilities for two runs of Subject S8. The first figure illustrates a typical result, while the second highlights a run that includes a misclassification of one airtime which is referenced/discussed in line 255, 365 and 421. 

Comment5: I would like to suggest including the gyroscope data into the model input and comparing the performance by input data in the future study.

Response5: We added a notion on additional input for the network to the Discussion (line 403-404).

Comment6: “3.2. Measurement system”  “3.2. Measurement System”

Response6: Done

Reviewer 2 Report

Comments and Suggestions for Authors

This article presents a novel and interesting application of machine learning algorithms in snowboard halfpipe, making it an enjoyable read. The effort and dedication of the authors are clearly evident throughout. However, despite the substantial research, the article could benefit from further clarification and improvements. I offer the following suggestions to help clarify and refine the manuscript for publication in this journal. I believe the authors should have no difficulty addressing these points:

1. I completely understand that this special study population is very hard to recruit many subjects to achieve enough study power overall. My major concern is that how the proposed machine learning approach can overcome a potential underpower of the study. Moreover, looking at overview of subject characteristics, the number of runs and hits were quite different, especially in the subject S4 and S5, indicating substantial heterogeneity across subjects. Can the authors address this issue and how they accommodate such heterogeneity in machine learning method?

2. Could the authors clarify the selection of the distribution and the hyper-parameters in Table 2? I don’t see any rationale for this justification or why they chose? Considering the small number of sample size in the study, I strongly recommend the authors to include a subsection of sensitivity analysis given the distribution and hyper-parameters? These findings and observations should be discussed in Discussion Section.

3. Table 3 delivered the results of the experiments by the usage of different percentage in training samples. Of course, 100% train samples gained lots of improvement. However, the Hausdorff distance only improved 2.3 in 50% compared to 2.2 improvement in 20% case when left setting. Moreover, the Hausdorff distance of left/right using 50% case is less than that of 100%. Is it correct information to report? I was not impressed by such very minor improvement, so what is the main message that the authors would like to emphasize? How can potential readers understand and interpret on this result?

4. The results seem restricted to Hausdorff distance. I strongly recommend that the authors to explore other distance measure to ensure robustness of the results. If different distance metrics provide substantial discrepancies, this should be discussed in details in the Discussion Section. The authors should explore why such discrepancies occurred and suggest what distance measure would be the most appropriate for the proposed methodology.

Comments on the Quality of English Language

Writing needs to be more polished.

Author Response

Dear reviewer,

We would like to take this opportunity to thank all reviewers for their comprehensive and constructive reviews. We are very pleased that the reviewer explicitly supports our aim to investigate airtime detection via machine learning methods in more detail. The comments to the reviewers' are given below and the changes to the manuscript are highlighted in yellow.

Comments1: 

I completely understand that this special study population is very hard to recruit many subjects to achieve enough study power overall. My major concern is that how the proposed machine learning approach can overcome a potential underpower of the study. Moreover, looking at overview of subject characteristics, the number of runs and hits were quite different, especially in the subject S4 and S5, indicating substantial heterogeneity across subjects. Can the authors address this issue and how they accommodate such heterogeneity in machine learning method?

Response1: It is true that recruiting a large number of subjects for this specific high-performance setup is challenging. However, we managed to recruit twice as many athletes compared to the baseline study and applied a rolling window approach to our time-series data. This allowed us to generate additional training samples from the available data, enhancing the model’s ability to capture more patterns and variability in each athlete's performance. To ensure the models are not prone to biases due to heterogeneity in the number of tricks/runs between athletes, each athlete has the same number of runs in the validation and test data. Regarding the training phase, while some athletes had more training data available, this primarily allowed the model to learn from a broader range of variability during training without impacting the fairness of validation and test evaluations. Furthermore, as shown in our analysis (Table 5), the model generalizes well even to unseen subjects, indicating that the approach successfully accommodates the inherent variability in subject characteristics. Nevertheless, we recognize that specific characteristics of different subjects, tricks, conditions, or locations could be influential and should be mitigated by providing richer datasets, as demonstrated by the data reduction (Table 3). This is also why we included an analysis of the improvements through fine-tuning to athlete-specific data, concluding that this approach is promising for compensating for potential limitations in the dataset (see section "U-Net Superiority and the Option for Balanced Fine-Tuning" in the discussion - line 418-439).

Comments2: Could the authors clarify the selection of the distribution and the hyper-parameters in Table 2? I don’t see any rationale for this justification or why they chose?

Response2: 

  • Our U-net approach is based on Zhang, Y.; Zhang, Z.; Zhang, Y.; Bao, J.; Zhang, Y.; Deng, H. Human activity recognition based on motion sensor using u-net. IEEE Access 2019, 7, 75213–75226. https://doi.org/10.1109/ACCESS.2019.2920969. In our opinion, the most relevant parameters to optimize are learning rate, dropout and units which is why we chose to focus on those Parameters.
  • We used common ranges for optimization and stated “The fact that the determined optimal hyperparameters do not represent extreme values within the specified parameter range confirms the appropriateness of the chosen ranges for tuning and suggests that the models are well-optimized based on the given data.” (line 393-395)
  • Window steps were set arbitrarily for augmentation
  • window size was set for 2 seconds to capture at least one full jump (“considering the capture frequency of 201.03 Hz, a window size of 400 frames will create data windows including the complete jump featuring take-off and landing even for the maximum airtime of 1.91 s.” [line 220-222]).
  • After double checking: in table 2 we corrected loguniform distribution to uniform

Comment3: Considering the small number of sample size in the study, I strongly recommend the authors to include a subsection of sensitivity analysis given the distribution and hyper-parameters? These findings and observations should be discussed in Discussion Section.

Response3: To give more insights into the hyperparameter tuning and impact of different configurations on our metrics we added Parralel line charts from Weights and Biases where we tracked our Algorithm training (Appendix A) and referred to this in line 279-280.

Comment4: Table 3 delivered the results of the experiments by the usage of different percentage in training samples. Of course, 100% train samples gained lots of improvement. However, the Hausdorff distance only improved 2.3 in 50% compared to 2.2 improvement in 20% case when left setting.

Response4: The improvements stated here seem to be calculated as a difference between validation and test at the specific level. However, the improvements should be compared between different rows of the same column. The difference in the test (left) is actually 0.864 in 50% and additional 0.801 in 20% (1.665 from 20% to 100%).

Comment5: Moreover, the Hausdorff distance of left/right using 50% case is less than that of 100%. Is it correct information to report?

Response5: This is correct but can be seen as a statistical artifact in the validation set, in which the model was luckily very good for these jumps. However, comparing the test set, which is the relevant one for assessing model generalization, the expected improvement is evident.

Comment6: I was not impressed by such very minor improvement, so what is the main message that the authors would like to emphasize? How can potential readers understand and interpret on this result?

Response6: This study was designed to show the tradeoff between labeling effort and prediction quality. We agree that the improvements seem small, but might be worth the effort depending on the use case (using only 20% of the data lead to a 28.58% worse Hausdorff distance for the combined sensor setup and to a 20.59% worse Hausdorff distance for the left sensor setup when compared to 100% data). For example in a competitive training environment, every ms of a correct timing of the athlete’s movement is important to gain insights into the biomechanics of the crucial time points of takeoff and landing which is already addressed in the manuscript: “These advancements facilitate real-time feedback and detailed biomechanical analysis, enhancing performance and trick execution, particularly during critical events such as take-off and landing, where precise time-domain localization is crucial for providing accurate feedback to coaches and athletes.” (line 16-19)

Comment7: The results seem restricted to Hausdorff distance. I strongly recommend that the authors to explore other distance measure to ensure robustness of the results. If different distance metrics provide substantial discrepancies, this should be discussed in details in the Discussion Section. The authors should explore why such discrepancies occurred and suggest what distance measure would be the most appropriate for the proposed methodology.

Response7: As stated in section 3.5.1 we used the Binary IoU as a proxy metric for early stopping for easy calculation. The best Hausdorff overall corresponds to the best Binary IoU which is shown by the added parallel lines charts from the hyperparameter tuning in appendix A. For reporting we focused on the Hausdorff, as it compares the worst assignment of a changepoint, while the IoU does not care for the position of the error, just the number of errors.

Round 2

Reviewer 2 Report

Comments and Suggestions for Authors

I appreciate the authors response, but still there are more clarifications before publishing this journal.

1. I don’t think that the current response from the authors is an appropriate to address heterogeneity across subjects. Since the machine learning approach is optimization-based estimates, the model cannot be generalized to the target population. It will be different estimates and results if the authors recruit more subjects in the model. Due to this nature, the authors response “..each athlete has the same number of runs in the validation and test data” is not correct. Basically, more observations are measured by each subject, there will be more variabilities and correlation. I recommend the authors to calculate I2 statistics, representing heterogeneity across subjects. As the airtime for each subject (S1 to S8) is already presented in Table 1, it is very trial to calculate it.

2. On my previous comment on sensitivity analysis given the small number of sample sizes, the authors provided line charts (Figures A1 to A6) in Appendix A. However, this approach alone is not helpful for the potential readers without further explanation by the author’s findings. Can the authors add more text on their observations/findings and potential issue related to this sensitivity analysis in Discussion Section? Whatever the author includes in the revision, they should be able to provide some messages. Otherwise, it should not be in the manuscript.

Author Response

Comments1: I don’t think that the current response from the authors is an appropriate to address heterogeneity across subjects. Since the machine learning approach is optimization-based estimates, the model cannot be generalized to the target population. It will be different estimates and results if the authors recruit more subjects in the model. Due to this nature, the authors response “..each athlete has the same number of runs in the validation and test data” is not correct. Basically, more observations are measured by each subject, there will be more variabilities and correlation. I recommend the authors to calculate I2 statistics, representing heterogeneity across subjects. As the airtime for each subject (S1 to S8) is already presented in Table 1, it is very trial to calculate it.

Response1: 

We would like to thank the reviewer for the valuable suggestions regarding the heterogeneity of the data foundation. We agree that there is considerable heterogeneity among the different subjects. These differences reflect the individual performances and specific characteristics of each athlete, which is, to some extent, to be expected. However, we would like to highlight several aspects of the present manuscript that should be considered to provide a comprehensive understanding of the impact of this heterogeneity and its potential effects on our proposed detection algorithm:

  1. I² Statistic Calculation: As recommended by the reviewer, we calculated the I² statistic to investigate the heterogeneity of the airtimes, resulting in a value of 99.57%. This value indicates substantial heterogeneity, which we do not dispute.
  2. Model Robustness and Generalization: To enhance the robustness of the model and prevent overfitting, we implemented an early stopping algorithm (with a patience of five epochs) using the Binary Intersection over Union (IoU) metric. This ensures that the model does not become overly tailored to specific training data, thereby improving generalizability (see lines 171-173).
  • Additionally, we would like to highlight the results in Table 5, which demonstrate that even in a zero-shot learning scenario (i.e., applying the model to an unseen athlete without fine-tuning), our model outperforms the baseline by a considerable margin. This indicates that our model is not exclusively tuned to the specific characteristics of the training subjects but can also be effectively applied to new subjects.
  1. Subject-Specific Fine-Tuning: Recognizing the limitations mentioned, we propose subject-specific fine-tuning as a further improvement step (see lines 420-441). This approach could enable the model to better account for individual differences between athletes and further improve prediction accuracy. Our model serves as a foundation for understanding the various IMU characteristics and their relation to airtime or ground contact. Through training from scratch, we demonstrated that our model provides a solid and generalized foundation for fine-tuning (see lines 359-360). The fine-tuning was able, within just a few epochs, to achieve results reaching a similar range as those of Subjects 1-7 (see Table4).
  2. Cross-Validation Approach: In future studies, we would suggest using a broader dataset to enable a leave-one-subject-out cross-validation (k-fold) approach. Unfortunately, due to the constraints of our dataset, this was not feasible in the current study. Specifically, considering the two measurement locations, one setting with only two subjects (measured at Kitzsteinhorn) did not allow for such a split, while the other setting resulted in a very limited training data foundation. To clarify this to the reader, we added two short paragraphs: one in Section 3.7.1. (line 214-216) explaining the chosen leave-one-run-out strategy, and another in the outlook (line 454-456) highlighting the potential of more data and a k-fold cross-validation approach.

Comments2: On my previous comment on sensitivity analysis given the small number of sample sizes, the authors provided line charts (Figures A1 to A6) in Appendix A. However, this approach alone is not helpful for the potential readers without further explanation by the author’s findings. Can the authors add more text on their observations/findings and potential issue related to this sensitivity analysis in Discussion Section? Whatever the author includes in the revision, they should be able to provide some messages. Otherwise, it should not be in the manuscript.

Response2: 

This is a good point. We added some clarifications for the reader in the Appendix A:
"The figures show that hyperparameters such as dropout rate, number of units, and learning rate significantly impact the model’s performance. Specifically, higher dropout rates generally led to lower performance, indicating a risk of underfitting when the regularization effect was too strong. The number of units also played a key role, where too few units resulted in reduced feature extraction capability, affecting overall accuracy. The learning rate demonstrated a trade-off between stability and convergence, with higher rates leading to performance fluctuations, while lower rates provided stability but slower convergence. Both IoU and Hausdorff distance metrics showed similar performance trends across the various hyperparameter settings, suggesting that IoU effectively served as a metric for early stopping."